# ComK2 represses competence development for natural transformation in Staphylococcus aureus grown under strong oxygen limitation

Shi Yuan Feng[1,3], Yannis Arab [1,2,3], Yolande Hauck[1], Pierre Poirette[1,2], Magali Noiray[1],
Sophie Quevillon-Cheruel [1], Stéphanie Marsin [1], Jessica Andreani [1] & Nicolas Mirouze [1,2] ✉

The facultative anaerobe and major human pathogen *Staphylococcus aureus* is able to sustain growth under a wide range of oxygen concentrations. Importantly, we have already demonstrated that under microaerobic conditions, sensed by the two-component system SrrAB, *S. aureus* efficiently induces the development of competence for natural transformation, one of the three main horizontal gene transfer mechanisms present in bacteria. Here, we show that when the oxygen concentration decreases even further (reaching almost anaerobic conditions) the development of competence for natural transformation is still allowed but with much less efficiency than under microaerobic conditions. This inhibition is controlled by a central competence regulator, named ComK2, that was not found involved under intermediate oxygen concentrations. This ComK2-dependent inhibitory pathway also involves the SA2107 protein, of unknown function, through a direct protein-protein interaction. Finally, we demonstrate that this inhibition of competence is controlled by this strong oxygen limitation, sensed by another two-component system named NreBC, probably involved in the same pathway as ComK2 and SA2107. All in all, our results show that the oxygen concentration, which varies drastically depending on the site in the human body but also during bacterial infections, is a key environmental factor that tightly modulates *S. aureus* genomic plasticity.

*Staphylococcus aureus* is an asymptomatic and permanent colonizer of the human population but is also considered as one of the foremost opportunistic bacterial pathogens of humans. *S. aureus* colonizes up to 30% of humans in the nose and frequently in other sites such as the skin, throat, axillae, groin and intestine[1]. Colonization is usually harmless but can occasionally lead to a wide range of infections including mild skin infections but also deadly invasive complications, such as osteomyelitis, septic arthritis, septicemia, pneumonia and endocarditis[2]. The diversity of colonization and infection sites, as well as the interplay with the host defenses and microbiota is only possible because of an important number of genes and pathways allowing the impressive adaptability of *S. aureus*[3].

An important and fluctuating parameter that bacteria have to adapt to is the oxygen ($O_2$) concentration within the host[4]. Indeed, the amount of dissolved $O_2$ within the host tissues and cells depends on several factors including: barometric pressure, climatological

conditions (temperature, relative humidity, latitude, altitude), as well as physiological, pathological, and physical-chemical processes within the organism itself[5]. In the presence of $O_2$, aerobic bacterial species can conduct aerobic respiration during which $O_2$ acts as the final electron acceptor for the electron transport chain. In the absence of $O_2$, anaerobic bacteria use alternative metabolic pathways including anaerobic respiration and/or fermentation[6].

As a facultative anaerobic organism, *S. aureus* is able to sustain growth under a wide range of $O_2$ concentrations[7,8]. This ability is particularly important to promote an infection, on the onset of inflammation or in biofilm communities where microaerobic or anaerobic conditions are often encountered[9]. The switch from aerobic to anaerobic metabolisms in *S. aureus* is complex and highly regulated. The ability of *S. aureus* to adapt to extreme changes in external $O_2$ concentration implies the presence of several $O_2$-sensing systems (i.e. the SrrAB, NreBC and AirSR two-component

[1]Université Paris-Saclay, CEA, CNRS, Institute for Integrative Biology of the Cell (I2BC), Gif-sur-Yvette, France. [2]Université Paris-Saclay, UVSQ, Inserm, Infection et inflammation, Montigny-Le-Bretonneux, France. [3]These authors contributed equally: Shi Yuan Feng, Yannis Arab. ✉e-mail: nicolas.mirouze@inserm.fr

systems, TCS[10–12]) that regulate the expression of genes involved during the transition from aerobic to anaerobic growth.

Importantly, we have recently shown that *S. aureus* is able to sense a decrease of the $O_2$ concentration (leading to microaerobic conditions) and in response, optimally induces the development of a specific physiological state named competence and essential for natural transformation, one of the three main horizontal gene transfer mechanisms present in bacteria[13]. We hypothesized that similarly to other important model organisms, *S. aureus* induces competence for natural transformation in order to promote its genomic plasticity in stressing environments[13].

Three main central competence regulators have been identified and characterized in *S. aureus*[14,15]: a secondary sigma factor, SigH (comparable to ComX in *Streptococcus pneumoniae*[16]), and two genes, *comK1* and *comK2*, encoding putative homologs of the transcription factor ComK from *Bacillus subtilis*[17]. Among these three central competence regulators, we have demonstrated that only SigH and ComK1 are essential for the expression of the natural transformation genes under microaerobic conditions in CS2 medium[13]. We have then shown that the SrrAB TCS is important to activate SigH in response to low $O_2$ concentrations[13]. Moreover, while ComK2 was not found as an activator of the natural transformation genes expression under microaerobic conditions, it was still considered as a central competence regulator[13]. Indeed, its regulon encompasses genes involved in competence-associated secondary functions such as the general stress response, toxin-antitoxin systems or amino and nucleic acids metabolisms[13].

Moreover, the development of competence has not only been shown under microaerobic conditions but also in static and $O_2$-deprived cultures (i.e. anaerobic conditions)[18]. Indeed, Morikawa and colleagues have developed a synthetic medium, called GS, in which *S. aureus* is also able to develop competence under strong $O_2$ limitation[18]. Even though no correlation between the intensity of competence development and the $O_2$ availability has been documented yet, this capacity to induce horizontal gene transfer under a wide variety of conditions clearly demonstrates *S. aureus* adaptability.

Interestingly, it has been proposed that the use of multiple regulators might have evolved to allow *S. aureus* to use a wider range of cues to decide whether or not to become competent for natural transformation[13,15]. Indeed, in this study we confirmed that *S. aureus* is able to induce competence under stronger $O_2$ limitations even though much less efficiently than under microaerobic conditions. We particularly showed that while SigH and ComK1 are still activators of the natural transformation genes expression, the third central competence regulator, ComK2, represses the expression of some of these genes explaining the lower probability of inducing competence under strong $O_2$ limitation. We also identified a protein of unknown function, named SA2107, interacting with ComK2 and potentializing this repressor function. Finally, we found that the natural transformation genes repression is controlled by the NreBC TCS in response to $O_2$ rarefaction, acting in the same pathway as ComK2 and SA2107. Ultimately, we propose a model in which *S. aureus* modulates the development of competence, depending on the $O_2$ concentration, the stress imposed to the bacteria and the energy necessary to adapt.

## Results

### Competence development is allowed, but restricted, under strong oxygen limitation

To monitor the development of competence under strong $O_2$ limitation in GS medium, we optimized the protocol published by Morikawa and his colleagues in 2012[18], taking profit of our previous experience with CS2 medium[13] (see material and methods for details). In order to compare our results, we used the same reporter strain as in[13], expressing the gfp gene under the control of the com*G* late competence operon's promoter ($P_{comG}$-gfp). Briefly, this reporter strain was first streaked on a BHI agarose plate. Isolated colonies are then used to inoculate a pre-culture in BHI medium. This pre-culture was then quickly stopped in exponential growth,

centrifuged, washed and used to inoculate a fresh culture in GS medium incubated statically at 37 °C. Finally, cell density (OD600nm), the percentage of GFP-expressing cells as well as the $O_2$ concentration were measured every 30 min.

Figure 1a shows how the $O_2$ concentration first dropped from 21 to 0.03% during the first 15 h of growth and even lower in the second half of the experiment (compare to the 0.3% measured under microaerobic conditions, Supp. Fig. 1 and [13]). As a consequence, we observed two distinct phases in the growth curve. Indeed, the growth rate of the culture was lower once the $O_2$ concentration decreased below 15%, probably reflecting the metabolic adaptation of the cells to the less energetically efficient anaerobic metabolism. In addition, when the $O_2$ concentration decreased below 8%, the expression of the *comG* operon was authorized, to finally reach around 5% of the total population after 26 h of growth in GS medium. This result is 10 times lower than what we obtained under microaerobic conditions[13], suggesting that even though competence is allowed, an inhibitory mechanism must be present to limit its development under stronger $O_2$ rarefaction.

Accordingly, the transformation efficiency of a wild type strain grown in GS medium was evaluated around $2.1 \times 10^{-7}$ ($\pm 3.7 \times 10^{-7}$), a result 20 times lower than under microaerobic conditions (i.e. $4.2 \times 10^{-6}$, [13]) (Fig. 1b). Despite this concordance of results, it is important to mention that our natural transformation protocol presents an experimental limit. Indeed, to provide the exogenous DNA allowing the selection of transformants, we have to open the tubes and therefore disrupt the $O_2$ limitation. Therefore, we cannot exclude the fact that some of the transformants appeared after the exogenous DNA was added under aerobic conditions. If that is the case, it means that we underestimate the observed repression of competence development occurring under anaerobic conditions.

### SigH and ComK1 also activate the expression of the transformation genes under strong oxygen limitation

Next, we decided to test the effect of the deletion of the genes encoding the three central competence regulators, namely *sigH*, *comK1* and *comK2*, on the expression detected from the promoter of two genes or operons representative of the two natural transformation genes classes: $P_{comG}$ (a class I promoter controlled by both SigH and ComK1 under microaerobic conditions[13]), and $P_{ssb}$ (a class II promoter exclusively controlled by ComK1[13]). Interestingly, the specific activation of class I and II natural transformation genes is still present under strong $O_2$ limitation. Indeed, when *sigH* was deleted, only the expression from $P_{comG}$ was lost (Fig. 2a, b) while the expression from both $P_{comG}$ and $P_{ssb}$ was abolished when *comK1* was inactivated (Fig. 2a, b). Consistently with these results, the absence of both *sigH* or *comK1* abolished genetic transformation under strong $O_2$ limitation (Fig. 1b).

### ComK2 inhibits the expression of the *comG* operon under strong oxygen limitation

Importantly, while no effect on the expression of the natural transformation genes could be associated to the deletion of *comK2* under microaerobic conditions[13], we found that the absence of ComK2 led to a 4-fold increase of the expression from the $P_{comG}$ promoter under stronger $O_2$ limitation (Fig. 2a). Accordingly, we showed that deletion of *comK2* led to an increase of the natural transformation efficiency reaching $5.4 \times 10^{-6}$ ($\pm 7.8 \times 10^{-6}$) (Fig. 1b), consistent with the difference observed in the percentage of cells expressing the *comG* operon between *comK2* and wild type strains (Fig. 2a). This result suggests that ComK2 might be a repressor of the *comG* operon expression, limiting the development of competence and hence natural transformation under strong $O_2$ limitation. Furthermore, this ComK2 repression could not be confirmed with the $P_{ssb}$ promoter (Fig. 2b), potentially indicating a specific mode of action to regulate the *comG* operon expression.

### Yeast two-hybrid reveals a potential interaction between ComK2 and SA2107

Because ComK2 is present in both microaerobic and stronger $O_2$ limitation but the ComK2-associated inhibition of the *comG* operon is only detected in

Fig. 1 | Competence development under strong $O_2$ limitation in GS medium. a A wild-type strain (St29) expressing GFP under the control of $P_{comG}$ was grown in a GS medium for 28 h. The oxygen concentration, the percentage of GFP-expressing cells, and OD600 nm were measured every 30 min. Results are presented as mean ± SD. Each experiment has been repeated three times (biological replicates). b Transformation efficiencies of wild type (N315ex w/o Phi), *sigH* (St45), *comK1* (St37), *comK2* (St38), *sa2107* (St39), and *nreC* (St118) mutant strains using chromosomal DNA (gray bars) (see Material and methods for details). Results are presented as mean ± SD. For each strain, the experiment was repeated at least eight times (biological replicates). Individual experiments are shown as blue circles. Statistical significance was evaluated by two-way ANOVA with Tukey's post-hoc test as follows: ***$P < 0.001$ and *$P < 0.05$.

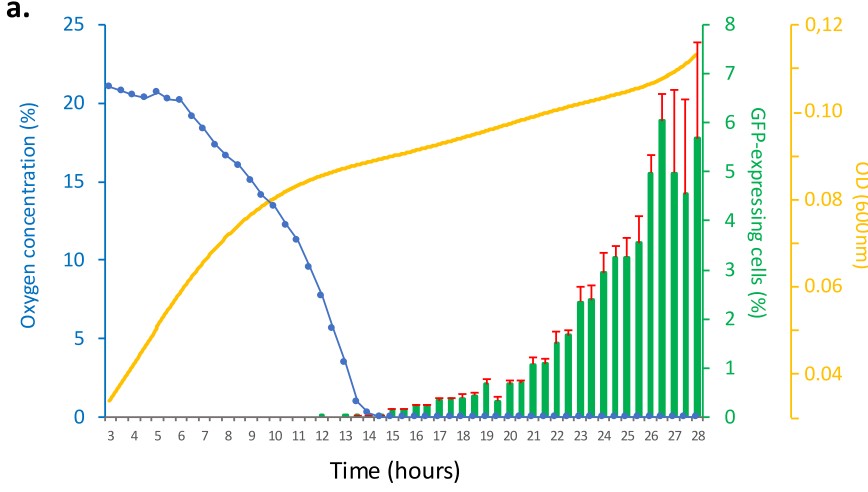

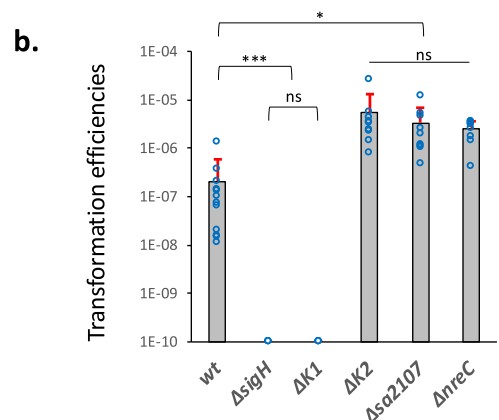

the latter, we first hypothesized that ComK2 needed to be potentialized by interacting with a partner. Interestingly, we identified an interaction between ComK2 and a protein of unknown function, SA2107, obtained in a yeast two-hybrid experiment performed as described previously ([19], Fig. 3a). Briefly, a fusion between ComK2 and the GAL4 binding domain (BD) was used as bait to screen a *S. aureus* genomic library constructed in a GAL4 activation domain (AD) prey vector (see Material and methods). A fragment corresponding to the full length SA2107 protein was identified in this screen (Fig. 3a).

### Structural modelling of the ComK2/SA2107 interaction

In addition, we used AlphaFold2 to further study the interaction between ComK2 and SA2107. First, the models proposed by AlphaFold2 for the individual proteins are robust with a pLDDT value superior to 70 for SA2107 and even 90 for ComK2 along the entire length of the proteins (Supp. Fig. 2). ComK2 is shown as a protein composed of 3 α-helices and 9 β-sheets while SA2107 is a short protein only displaying 4 α-helices (Supplementary Fig. 2). Then, Fig. 3b shows the best model generated by AlphaFold2 of the potential interaction between ComK2 and SA2107. Importantly, the predicted model for the complex formed by ComK2 and SA2107 seems very strong as the ipTM score (interface pTM) is superior to 0.8 (on a range from 0 to 1) (Supplementary Table 1). It is usually accepted that AlphaFold2 models of interactions displaying an ipTM superior to 0.7 can be considered as a strong starting point. Using the PyMol software, we next looked for the amino acids potentially involved in the ComK2-SA2107 interaction (Supplementary Table 2). We identified 8 hydrogen bonds as

well as 1 salt bridge involving amino acids present in ComK2 and SA2107 that are less than 4 Å apart.

To further challenge our confidence in this model, as well as the specificity of this interaction, we then evaluated the ability of SA2107 to interact with ComK1. Indeed, in addition of displaying around 27% of sequence similarity, ComK1 and ComK2 predicted structures are very close (Supplementary Fig. 3). The two proteins are almost perfectly superposable apart from N-terminal extra amino acids and an additional α-helix present in the C-ter of ComK1 (Supplementary Fig. 3). However, despite this strong homology, AlphaFold2 revealed that the interaction between ComK1 and SA2107 was very unlikely as the ipTM and combined pTM score for these models were below 0.3 (Supplementary Table 3). This difference could be explained by the N- and C-terminal extremities of ComK1 that could prevent this interaction and ultimately underline the specificity of the ComK2-SA2107 interaction (Supplementary Fig. 3).

### The interaction between ComK2 and SA2107 is binary and direct

In order to confirm that the interaction between ComK2 and SA2107 is binary and direct, we then purified the two recombinant His-tagged proteins (Supplementary Fig. 4). We then used Flow-Induced Dispersion Analysis (FIDA[20]) to investigate the ability of these proteins to interact. By measuring the fluorescence of a ligand in a laminar flow and analyzing its dispersion over time, FIDA allows the calculation of the ligand's apparent hydrodynamic radius ($R_h$). Furthermore, when mixed with a potential interactant, this technique measures the change in size of the ligand ($R_{hi}$) as it selectively interacts with the target protein under native conditions. Interestingly, the

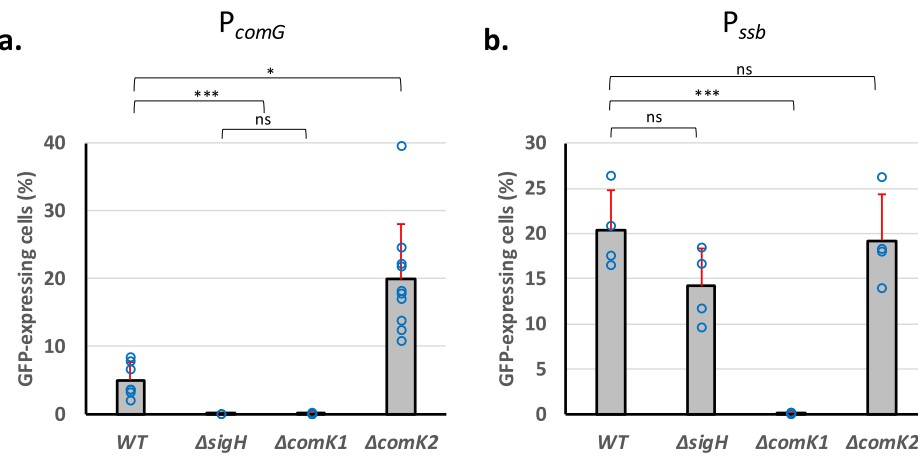

**Fig. 2 | ComK2 inhibits competence and natural transformation under strong O₂ limitation.** The percentage of the population expressing GFP under control of P$_{comG}$ (St29, St51, St40, and St 41) (**a**) or P$_{ssb}$ (St50, St61, St64, and St67) (**b**) in a wild-type background or in the absence of *sigH*, *comK1*, or *comK2* was determined after 26 h of growth in GS medium. Results are presented as mean ± SD. Each experiment has been repeated at least four times (biological replicates). Individual experiments are shown as blue circles. Statistical significance was evaluated by two-way ANOVA with Tukey's post-hoc test as follows: ***$P < 0.001$ and *$P < 0.05$.

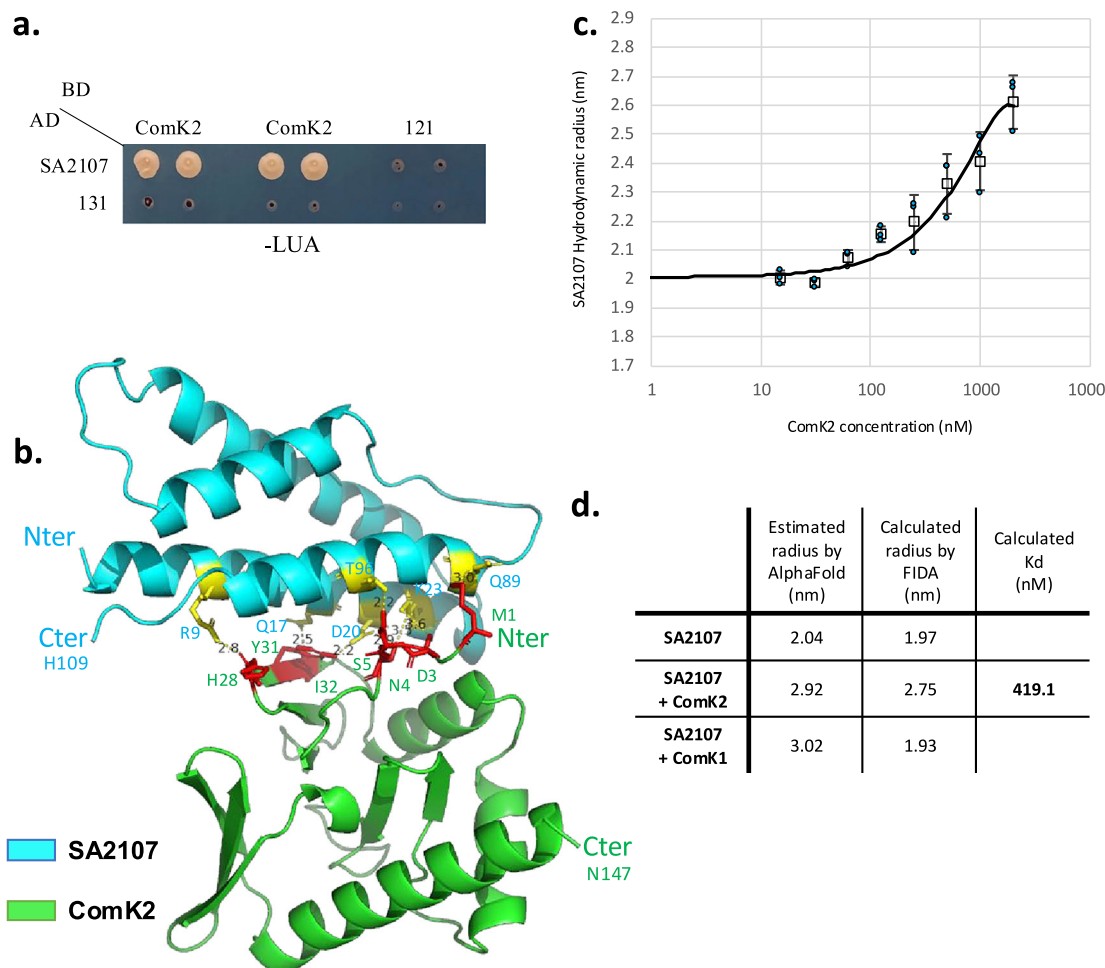

**Fig. 3 | ComK2 interacts with SA2107. a** Yeast two-hybrid experiment revealing a physical interaction between ComK2 and SA2107. A fusion between ComK2 and the GAL4 binding domain (BD) was used as a bait to screen a *S. aureus* genomic library constructed in a GAL4 activation domain (AD) prey vector. The full length SA2107 protein was identified in this screen. The interaction is revealed on a complete, synthetic and selective medium lacking leucine, uracile and adenine (to select for expression of the ADE2 interaction reporter). Negative controls harboring empty vectors (empty pBGDU, strain 121 and empty pGAD, strain 131). **b** Predicted structure of the ComK2-SA2107 complex using AlphaFold2. ComK2 is displayed in green and SA2107 in cyan. The amino acids at the interaction interface are highlighted in the "wire format" in yellow for SA2107 and in red for ComK2 (visualization of residues as sticks, see Supp. Fig. 2 for details). **c** Measured Hydrodynamic radius of SA2107 in complex with ComK2 using the Flow Induced Dispersion Analysis (FIDA[26]) method. A constant concentration of labelled SA2107 (20 nM) is exposed to increasing concentrations of ComK2 (15 nM-2 μM). SA2107 hydrodynamic radius is calculated based on three independent experiments (mean and SD). For each data point, individual hydrodynamic radius values are presented as blue circles. **d** Comparison of the calculated and estimated hydrodynamic radius of SA2107 alone and in the presence of ComK2 or ComK1. The dissociation constant (Kd) of the ComK2-SA2107 interaction is shown.

dissociation constant (Kd) of the interaction may also be obtained in a titration experiment.

Here, we decided to fluorescently label SA2107, used as the indicator. When analyzed alone, FIDA measured SA2107 Rh around 1.97 ±0.02 nm (Fig. 3c, d). This result is very close from what we evaluated when using SA2107 AlphaFold structural models (i.e. $R_{ha}$ = 2,05 ±0.02 nm, Fig. 3d). We then mixed labelled SA2107 and unlabeled ComK2 in order to measure the change in size of the potential protein-protein complex. FIDA evaluated the SA2107-ComK2 $R_{hi}$ around 2,75 ±0.09 nm, which is consistent with the AlphaFold structural model of the complex ($R_{ha}$ = 2,81 ±0.09 nm, Fig. 3c, d). Furthermore, a titration experiment, in which SA2107 concentration was fixed at 20 nM and mixed with ComK2 at concentrations ranging from 15 nM to 2 μM, we evaluated the Kd of the interaction at 420 ± 130 nM (Fig. 3c, d).

Finally, we decided to demonstrate that the SA2107-ComK2 interaction was specific and that as predicted by AlphaFold ComK1 could not interact with SA2107. This is why we reproduced the experiments presented above using the purified His-tagged ComK1 protein as a potential interactant (Supplementary Fig. 4). As expected, SA2107 Rh did not significantly change in the presence of ComK1, even at the higher concentrations (Supplementary Fig. 5), proving that despite the structural similarities observed between ComK1 and ComK2, SA2107 specifically interacts with the latter.

### ComK2 and SA2107 inhibit *comG* operon expression

Finally, we wanted to verify if SA2107 could be involved, alongside ComK2, in the inhibition of the expression of natural transformation genes. This is why we investigated the impact of the *sa2107* gene deletion on the expression from the P*comG* promoter. Interestingly, the absence of SA2107 led to a 4-fold increase in the expression from the P*comG* promoter (Fig. 4). Accordingly, the transformation efficiency of a *sa2107* mutant strain was comparable to that of a *comk2* mutant strain and evaluated around $3.3 \times 10^{-6}$ ($\pm 3.6 \times 10^{-6}$) (Fig. 1b).

Since ComK2 and SA2107 interact and that the deletion of both genes lead to a similar increase in the *comG* operon expression, we then asked if these two proteins could work together to inhibit the development of competence. Accordingly, the combination of the two genes deletion led to similar activation of the expression from the P*comG* promoter as in the single mutant strains (Fig. 4). This last result strongly suggests that the two proteins act in the same regulatory pathway, probably through a direct protein-protein interaction, leading to the inhibition of the *comG* operon expression.

### Strong oxygen limitation is sensed by the two-component system NreBC to inhibit *comG* operon expression

We previously showed that microaerobic conditions were sensed by SrrAB, one of the three TCS sensing $O_2$ in *S. aureus*, to stimulate competence development[13]. Therefore, we wondered if $O_2$-sensing TCS were involved in the regulation of competence development under stronger oxygen limitation. This is why we tested the impact of the deletion of the gene encoding each $O_2$-sensing TCS' response regulator on the percentage of *comG*-expressing cells. Remarkably, a *srrA* mutant did not affect the development of competence under strong $O_2$ limitation (Fig. 5a). Similar results were obtained in a *airR* mutant strain (Fig. 5a). Strikingly, the percentage of *comG*-expressing cells was increased by a 4-fold factor in a *nreC* (encoding the TCS response regulator) mutant strain (Fig. 5a), a phenotype very close from what was obtained in *comK2* and *sa2107* mutant strains (Fig. 4b). This result was confirmed by the transformation efficiency of a *nreC* mutant strain (Fig. 1b), which we found higher than the wild type strain, reaching $2.5 \times 10^{-6}$ ($\pm 1.1 \times 10^{-6}$), and similar to that of *comK2* and *sa2107* mutant strains.

Next, we verified if the NreBC TCS could sense strong $O_2$ rarefaction and in response activate the ComK2/SA2107 inhibitory pathway. To do so, we evaluated the percentage of *comG*-expressing cells in a mutant strain lacking both *nreC* and *comK2*. Interestingly, this double mutant strain displayed the same phenotype (i.e. a 4-fold increase in the percentage of *comG*-expressing cells) as the single mutant strains (Fig. 5b). This result probably reflects the fact that NreBC, ComK2 and SA2107 act in the same pathway to inhibit the development of competence under strong $O_2$ limitation.

## Discussion

### Three central competence regulators control the expression of the NT genes in *S. aureus*

Three putative central competence regulators have been identified in the human pathogen *S. aureus*: the alternative sigma factor SigH and the two transcriptional regulators ComK1 and ComK2[15,18]. We previously showed that only SigH and ComK1 were essential for the expression of the genes involved in natural transformation under microaerobic conditions (Fig. 6 and ref. 13). Meanwhile, ComK2 was found important for the general competence transcriptional program by controlling the expression of genes involved in secondary functions, such as the general response to stress, toxin-antitoxin systems or the nucleic and amino acid metabolism[13]. Interestingly, we show here that ComK2 also displays the ability to inhibit the expression of the natural transformation genes under stronger $O_2$ limitation (Fig. 6).

Ultimately, we demonstrate the existence of three true central competence regulators, modulating natural transformation, making the development of competence in *S. aureus* particularly complex and intricated. Each regulator can be induced or modulated by independent pathways allowing a fine tuning of the competence transcriptional program, of the percentage of cells inducing competence and as a consequence of the global transformation efficiency.

To our knowledge, *S. aureus* is the only model organism that has three central competence regulators. Indeed, the two historic model organisms only display one central competence regulator named ComX in *S. pneumoniae*[16] and ComK in *B. subtilis*[17]. However, it has been shown that *Vibrio cholerae*, the causative agent of cholera epidemics, uses two central competence regulators (namely TfoX and HapR)[21]. It has been hypothesized that this multiplicity of central competence regulators could be used to finely tune the development of competence in response to changes in the environmental conditions. It seems that in *S. aureus*, the presence of three central competence regulators is dedicated, at least in part, to the modulation of the development of competence in response to the $O_2$ concentration.

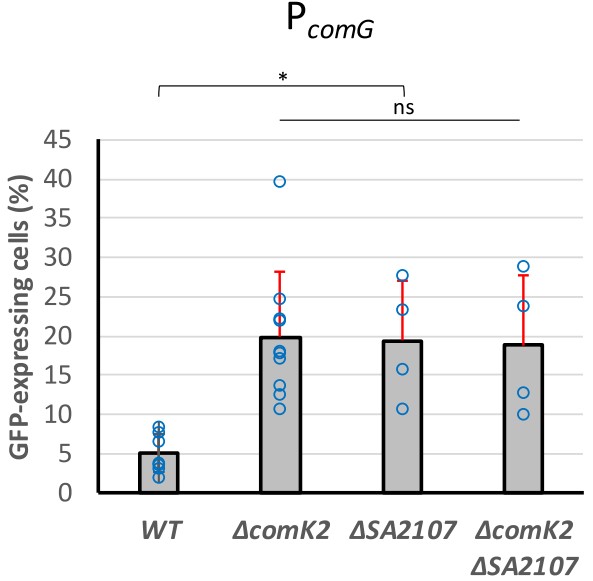

$P_{comG}$

**Fig. 4 | SA2107 also inhibits the P*comG* expression.** The percentage of the population expressing GFP under control of P*comG* (St29, St41, St42 and St 161) in a wild type background or in the absence of *comK2*, *sa2107*, or both, was determined after 26 hours of growth in GS medium. Results are presented as mean ± SD. Each experiment has been repeated at least 4 times (biological replicates). Individual experiments are shown as blue circles. Statistical significance was evaluated by two-way ANOVA with Tukey's posthoc test as follows: *$P < 0.05$.

**Fig. 5 | Inhibition of P$_{comG}$ expression is controlled the O$_2$-sensing two-component system NreBC. a** The percentage of the population expressing GFP under control of P$_{comG}$ (St29, St145, St158 and St177) in a wild type background or in the absence of *srrA*, *nreC*, or *airR* was determined after 26 hours of growth in GS medium. The results are presented as mean ± SD. For each strain, the percentage of the population expressing GFP has been calculated from at least 4 independent experiments (biological replicates). Individual experiments are shown as blue circles. **b** The percentage of the population expressing GFP under control of P$_{comG}$ (St29, St41, St158 and St273) in a wild type background or in the absence of *comK2*, *nreC*, or both, was determined after 26 hours of growth in GS medium. The results are presented as mean ± SD. For each strain, the percentage of the population expressing GFP has been calculated from at least 5 independent experiments (biological replicates). Individual experiments are shown as blue circles. Statistical significance was evaluated by two-way ANOVA with Tukey's posthoc test as follows: *$P < 0.05$.

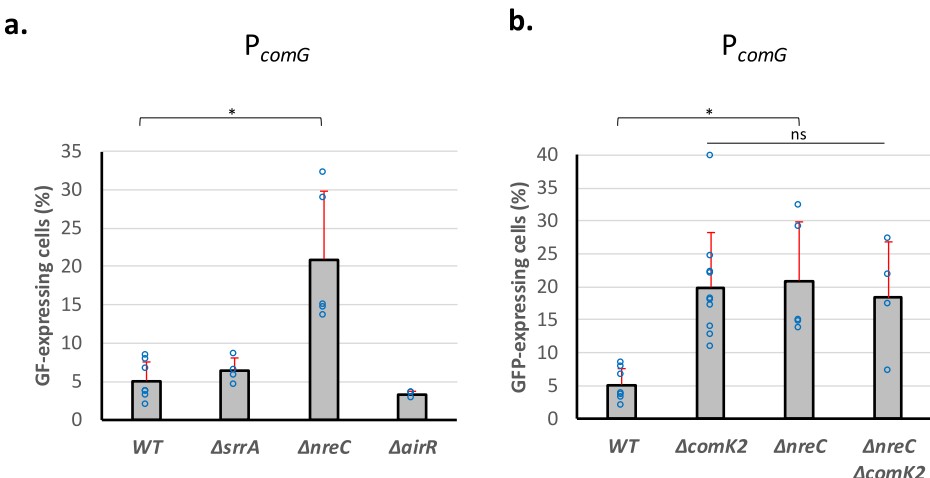

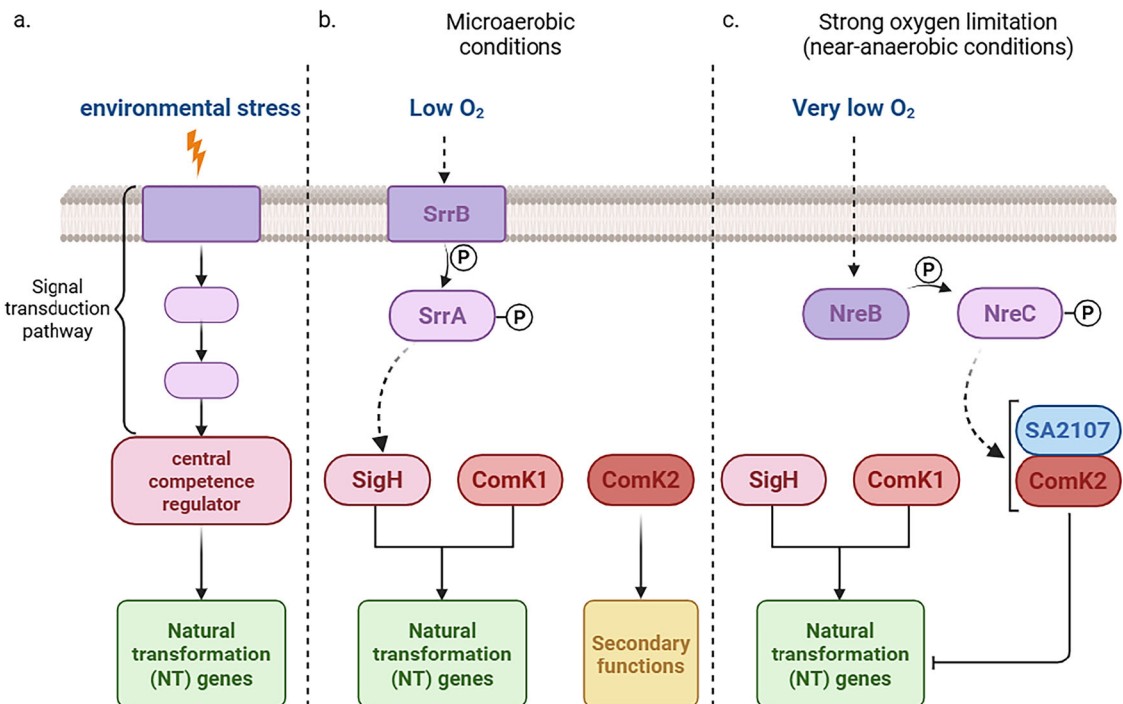

**Fig. 6 | Model for the regulation of the development of competence in *S. aureus* under strong O$_2$ limitation. a** The development of competence for natural transformation is usually presented as a sequential process. First, a signal (stress) is sensed by the cells, which in turn activate signal transduction pathways. These pathways ultimately activate the central competence regulators responsible for the activation of the late competence genes, among which are found all the genes encoding for the proteins involved in natural transformation. **b** Under microaerobic conditions, we have shown that oxygen rarefaction is sensed by the SrrAB two-component system which in turn activate the alternative sigma factor SigH. Together with ComK1, SigH is required to activate the expression of the natural transformation genes[13]. **c** Under strong oxygen limitation, SigH and ComK1 still activate the expression of the natural transformation genes but much less efficiently than under microaerobic conditions. This inhibition is performed by the third competence regulator, ComK2, probably through a direct protein-protein interaction with SA2107, a small protein of so far unknown structure and function. Furthermore, we showed that the two-component system NreBC senses the strong O$_2$ limitation and is involved in the ComK2/SA2107 inhibitory pathway, potentially through the activation of SA2107.

## Protein-protein interaction modulates activity of ComK2

We have shown that ComK2 activates the expression of competence genes involved in secondary functions in CS2 under microaerobic conditions[13]. However, we could not detect any inhibition of the natural transformation genes by ComK2 in these conditions[13]. Since ComK2 is present under both microaerobic and stronger O$_2$ limitation, but its inhibitory function is only present in the latter, we postulated that a co-repressor must be potentializing this new function. The identification of the interaction between ComK2 and SA2107 as well as the identical phenotypes of *comK2* and *sa2107* mutant strains led us to propose that SA2107 could be such co-repressor (Fig. 6).

Following the same reasoning that SA2107 is the key to promote this new ComK2-associated inhibitory effect, we can propose that SA2107 expression or modification, could be controlled, directly or indirectly, by the NreBC TCS in response to strong $O_2$ limitation (Fig. 6).

Interestingly, *sa2107* is only found in *S. aureus* genomes and does not display any annotated domain. However, when searching the AlphaFold database (AFDB) proteome with FoldSeek, we found the BssR protein (also called YliH) from *E. coli* with a very similar tridimensional predicted structure. BssR's name comes from the phrase "regulator of biofilm through signal secretion"[22]. Indeed, BssR is a small regulator involved in the inhibition of biofilm formation and the specific fold shared by BssR and SA2107 could represent a new family of small proteins, involved in the inhibition of bacterial environmental adaptations.

Finally, in order to explain how ComK2 and SA2107 could inhibit the expression of some natural transformation genes we can propose two hypotheses. First, the interaction between ComK2 and SA2107 could allow the complex to bind the promoter of natural transformation genes and prevent the access of SigH, ComK1 or the RNA-polymerase. However, preliminary results show that the ComK2-SA2107 complex cannot directly bind to the *comG* operon promoter (Supplementary Fig. 6). Alternatively, we can imagine that the complex formed by the two proteins is able to directly interact with SigH or ComK1, preventing the activation of the natural transformation genes expression through sequestration. Unfortunately, a strong unspecific binding of ComK1 to any DNA sequences prevented us to test the ability of the ComK2-SA2107 complex to inhibit ComK1 binding (Supplementary Fig. 6).

## Oxygen concentration tightly modulates competence development in *S. aureus*

Facultative anaerobes, such as *S. aureus*, constitute a unique class of bacteria able to grow in the presence or absence of $O_2$. On an evolutionary perspective, they can be considered as some of the most evolved and adapted bacteria for their ability to grow and disseminate within a wide range of microenvironments, especially during infection[23]. Therefore, it is not surprising to find in the list of priority and multi-resistant pathogens established by the WHO[24] an overrepresentation of facultative anaerobes. Upon closer inspection of the WHO's list, it has been established that a majority of species are known or suspected to be naturally transformable[25]. Therefore, it is striking to reveal that *S. aureus*, a facultative anaerobe, major human pathogen present in the WHO's list is able to modulate the development of competence for natural transformation, and its need of genomic plasticity and antibiotic resistances, depending on the $O_2$ concentration. Such correlation has already been proposed in *S. pneumoniae*[26], without a clear regulatory pathway proposed.

Investigations have shown that *S. aureus* finds itself limited for $O_2$ in vivo during the onset of infection or within biofilms. During bacterial infection, deprivation of $O_2$ in a tissue may result from inflammation that prevents blood from reaching the infection site which cannot keep pace with $O_2$ consumption from growing bacteria and recruited host immune cells[27]. Wound sites are example of tissues experiencing hypoxia and characterized by inflammation and bacterial infections[28]. Some diseases also create anoxic environments. This is the case of the sputum clogging the lungs of cystic fibrosis patients in which oxygen is depleted within the first few millimeters below the sputum-air interface[29]. As *S. aureus* is one of the main pathogens found in the lungs of young cystic fibrosis patients[30], this important human pathogen is clearly exposed, over time, to strong $O_2$ limitations. Finally, *S. aureus* is also known to grow microaerobically to anaerobically within biofilms which can lead to recurrent infections or septicemia[31]. Whether these biofilms are formed on medical devices or within the lungs of cystic fibrosis patients, they are all characterized by important gradients of $O_2$ limitations stressing the bacterial cells that need to adapt their metabolism. All these types of infections are so inherently linked to *S. aureus* pathogenicity that this human pathogen has to constantly adapt its energetic metabolism but also its need of genomic plasticity.

Ultimately, to explain such behavior, we can propose that even though *S. aureus* is able to grow under strong $O_2$ rarefaction, such environment remains stressing. Indeed, it has been shown that after a shift from aerobic to anaerobic growth (which occurs in our cultures), *S. aureus* growth rate was drastically reduced[8]. Therefore, it is plausible that stressed cells adapting to anaerobic conditions do not invest as much into competence for natural transformation, a well-known energy-consuming process. In addition, the low growth rate of *S. aureus* transformants under strong $O_2$ limitation could impact the potential gain of fitness associated to the new sequences acquired. Contrastingly, under microaerobic conditions, *S. aureus* cells could still use a combination of respiration, a less stressing and more energetically efficient metabolism, and fermentation, allowing the cells to invest more into competence for natural transformation[13].

## Materials and methods
### Bacterial strains and culture conditions

N315[18] *S. aureus* strains used in this project are all listed in Supplementary Table 4. *Staphylococcus aureus* strains were grown in BHI medium (Becton, Dickinson and Company) or a complete synthetic medium, called GS ([18], $K_2HPO_4$, 7 g/L; $KH_2PO_4$, 2 g/L; $Na_3$citrate · $2H_2O$, 0.4 g/L; $(NH_4)_2SO_4$, 1 g/L; $MgSO_4$, 0.05 g/L; Thiamine, 1 mg/L; Niacin, 1.2 mg/L; Biotin, 0.005 mg/L; D Pantothenate, 0.25 mg/L; Adenine, 5 mg/L, Guanine, 5 mg/L; Cytosine, 5 mg/L; Uracil, 5 mg/L; Thymine, 20 mg/L; L-alanine, 60 mg/L; L-arginine, 50 mg/L; L-aspartic acid, 90 mg/L; L-cystine, 20 mg/L; L-glutamic acid, 100 mg/L; L-histidine, 20 mg/L; L-isoleucine, 30 mg/L; L-leucine, 90 mg/L; L-lysine, 50 mg/L; L-methionine, 3 mg/L; L-phenylalanine, 40 mg/L; L-proline, 80 mg/L; L-threonine, 30 mg/L; L-tryptophan, 10 mg/L; L-tyrosine, 50 mg/L; L-valine, 80 mg/L) depending on the experiment. When necessary, antibiotics were used to select specific events (Kan, 200 µg/mL; Cm, 10 µg/mL).

### Protocol to naturally induce competence in *S. aureus* in GS medium

Cells were isolated from −80 °C stock on BHI plate. Four clones were inoculated in 10 mL of BHI (Becton, Dickinson and Company) and incubated at 37 °C with shaking at 180 rpm until OD reached 2.5. This pre-culture was then centrifuged, washed in fresh GS medium[18], and used to inoculate 2 mL of fresh GS medium (OD = 0.05) in closed 2 mL Eppendorf tubes. Several tubes of this culture were prepared in order to take individual samples along growth (i.e. each Eppendorf tube was only opened once for one sample). The cultures in GS medium were finally incubated overnight statically at 37 °C. Cells were collected throughout growth (GFP reporter strains, Fig. 1), after 16 h (transformation efficiencies, Fig. 2c) or 26 h (GFP reporter strains, Figs. 2, 4 and 5).

### Construction of *S. aureus* deletion mutants

In order to investigate the role of *sa2107* in the regulation of competence development in *S. aureus*, allelic replacement constructs were cloned into the temperature sensitive pIMAY plasmid[32] and used as presented in ref. 13. All the primers (IDT) used for cloning in the present study are listed in Supplementary Table 5.

### Flow cytometry to determine the percentage of the population expressing GFP

Following growth in GS, 4 mL of cells were harvested by centrifugation at 11,000 g for 4 min. Pellets were resuspended in 500 µL of cold 70% ethanol and incubated on ice for 30 min, in order to fix the cells. Then, *S. aureus* cells were resuspended in 500 µL of PBS (pH 7,4) after centrifugation at 11,000 g for 1 min. Finally, the percentage of the population expressing GFP was evaluated by Flow cytometry (Cytoflex top-bench cytometer, Beckman-Coulter). Following Forward- and Side-scatter detection to identify individual cells (20000 events), a 488 nm laser was used to distinguish GFP-expressing competent cells by comparison with the auto-fluorescence of a strain that did not express GFP (St12) (as already shown in ref. 13).

It is important to mention that GFP is a very stable protein. Therefore, once the maximum percentage of competent cells was reached, this number stayed constant for hours. This feature does not mean that competence stays 'open' for hours but rather that once the maximum is reached, no new competent cells appear.

## Natural transformation of competent *S. aureus* cells

Wild type strain (St12) as well as *sigH* (St45), *comK1* (St37), *comK2* (St38), *sa2107* (St39) and *nreC* (St118) mutant strains were first grown to competence in GS medium. Cells were naturally transformed following the protocol previously published[13].

**Transformation protocol.** Briefly, at each time point (every half hour), 8 mL of cells were harvested by centrifugation at 10,000 g for 4 min at 4 °C, resuspended in 500 μL of fresh CS2 (Note that at this stage the medium used to resuspend the cells contains normal oxygen concentrations). Five μg of donor- chromosomal DNA were added to one of the tubes (the second tube is used as a "no DNA" control) and incubated at 37 °C for 2.5 hours with agitation at 180 rpm. 50 or 500 μL (tube with DNA) or 1 ml ("no DNA control") from each tube where finally mixed with 25 mL of melted BHI (Becton, Dickinson and Company) agar pre-cooled to 55 °C together with antibiotic, and the mixture was poured into petri dishes. After solidification, the plates were incubated at 37 °C for 48 hours. At each time point, the viability was also evaluated by serial dilution on BHI (Becton, Dickinson and Company) agar plates. Transformation efficiencies were finally calculated by dividing the number of transformants detected in 1 mL of culture by the total number of cells in the same volume.

Numbers presented in Fig. 1C represent the mean of the highest transformation efficiencies detected along growth during each experiment. The experiments have been repeated for each strain at least 10 times to provide strong statistical relevance.

**Donor DNA preparation.** Chromosomal DNA: strain St294 was used to provide donor chromosomal DNA. In St294, the pIMAY-INT[15] plasmid (Cm) was inserted in the chromosome at the INT chromosomal site[15]. The plasmid insertion was verified by PCR while no replicating plasmid could be detected. Briefly, 100 mL of culture were centrifuged and resuspended in 5 mL of TEG (Tris 5 mM, pH8; EDTA, 10 mM; Glucose, 1%) complemented with 500 μL of Proteinase K (10 mg/mL, Invitrogen), 2 mL of lysis buffer (NaOH, 0.2 N; SDS, 1%) and 20 g of glass beads (Stratech, #11079-105, 0.5 mm in diameter). The cells were then broken using 5 cycles of vortex (1 min each) with 1 min in ice between each cycle. To finish cell lysis, 3 mL of lysis buffer were added for 5 min at room temperature and neutralized with 6 mL of NaAc (3 M, pH 4.8). Finally, chromosomal DNA present in the supernatant was precipitated using 96% ethanol (1 mL of EtOH for 500 μL of supernatant) after 2 hours of incubation at -20 °C. After centrifugation, chromosomal DNA was washed using 300 μL of cold 70% ethanol. Precipitated chromosomal DNA was finally resuspended in 300 μL of Tris-HCl 5 mM, pH8.

## Oxygen concentration measurements

Oxygen concentrations were measured using the SP-PSt3-SA23-D3-OIW oxygen sensor spots (PreSens GmbH, Regensburg, Germany). These self-adhesive sensor spots were attached to the inner wall of 2 mL Eppendorf tubes so that the spots would always be immerged during the experiments. The Eppendorf tubes were closed at T0 and remained closed for the entire experiment.

The sensor spots are covered with an oxygen-sensitive coating where molecular oxygen quenches the luminescence of an inert metal porphyrine complex immobilized in an oxygen-permeable matrix. This process guarantees a high temporal resolution and a measurement without drift or oxygen consumption.

The photoluminescence lifetime of the luminophore within the sensor spot was measured using a polymer optical fiber linked to an oxygen Meter (Fibox 4 trace; PreSens GmbH). Excitation light (505 nm) was supplied by a glass fiber, which also transported the emitted fluorescence signal (600 nm) back to the oxygen meter. Briefly, an oxygen measurement was realized, through the 2 mL Eppendorf tube plastic, by simply approaching the optical fiber from the sensor spot. At each time point, the oxygen concentration was measured three times and the results provided represent the mean of these three measurements. In our experiments, oxygen concentration was measured every 30 minutes.

## Yeast two-hybrid in yeast

The *comK2* gene cloned into a GAL4 BD (bait) vector was expressed in *Saccharomyces cerevisiae* to screen a *S. aureus* library essentially as described previously[33]. Briefly, the *S. aureus* library was constructed in a GAL4 AD (prey) vector in *E. coli* and transferred into yeast. Interactions were revealed by growth of diploid cells after 5 to 14 days at 30 °C on synthetic complete medium[34] lacking leucine, uracil, and histidine (to select for expression of the HIS3 interaction reporter) and further tested on synthetic complete medium lacking leucine, uracil and adenine (to select for expression of the ADE2 interaction reporter). Controls with empty vector plasmids (i.e., carrying only the BD or AD domain) were systematically included.

## Protein 3D structure prediction using AlphaFold2

AlphaFold2 is an artificial intelligence system developed by DeepMind capable of predicting the 3-dimensional structure of a protein only with its amino-acid sequence[35–37]. A user interface developed by the I2BC (host institute, https://bioi2.i2bc.paris-saclay.fr/tools/structural-biology/), facilitating the access to AlphaFold, has been used to predict the 3D structure of ComK1, ComK2 and SA2107, as well as their potential interaction. Additionally, the algorithm calculates different statistics quantifying the confidence in the predicted models. The pTM-score (represented by a number between 0 and 1) measures the accuracy of the entire structure, and it is accepted that a pTM-score above 0.5 means that the global predicted fold for the protein might be similar to the true structure. The ipTM-score (interface pTM-score) is a similar statistic, but only applied to the regions found at the interface between proteins in a complex. The "combined score" is a weighted sum of the pTM and ipTM and gives thus a better global estimation of the model's accuracy. In the same way as previously, a value close to 1 corresponds to a high confidence predicted model. Finally, the plDDT (predicted local distance difference test) corresponds to a measure of confidence in the interpretation of the structure, by amino acids. The PyMol software (The PyMOL Molecular Graphics System, Version 3.0 Schrödinger, LLC.) was used to generate images of different protein structures and to highlight the residues involved in the interactions. The PISA tool[38] (Proteins, Interfaces, Structures and Assemblies, EMBL-EBI) was then used to assign the amino-acids capable of forming hydrogen bonds, salt bridges, or hydrophobic interactions at the interface of proteins.

To compare the apparent hydrodynamic radius of SA2107 alone (Rh) or in a complex (Rhi) obtained through the FIDA method with estimated measurements, we used the AlphaFold2 predicted structures. The approximative diameter of 5 different models were obtained by tracing two lines covering the structures' maximum length using the "Measurement" tool of PyMol. Mean of these diameters were calculated, and divided by two to obtain the estimated hydrodynamic radius of SA2107, SA2107+ComK1 and SA2107+ComK2 (referred to as Rha).

## ComK1, ComK2 and SA2107 purification

After a PCR amplification of the *comK1*, *comK2* and *sa2107* genes from the St012 strain, using the NdeI-6His-comK1-F and XhoI-comK1-R, NdeI-comK2-F and Xho-comK2-His-R, and NdeI-His-SA2107-F and Xho-His-SA2107-R oligonucleotides respectively, followed by a cloning in pET21a (*comK2*) or pET29a (*comK1* and *sa2107*) vectors, a His6 tagged version of ComK1 and ComK2 were expressed in *Escherichia coli* BL21-Gold (DE3) or Rosetta (DE3) pLysS (Novagen) for SA2107. Protein expression was induced in 800 mL of cultures by using 2xYT medium supplemented with IPTG (100 μg/mL). After an overnight incubation at 15 °C, cells were

collected and the pellet resuspended in 80 mL of specific buffer (ComK1: 20 mM Phosphate buffer, 1 M NaCl, pH5.6; ComK2: 20 mM Phosphate buffer, 500 mM NaCl, pH5.6; SA2107: 50 mM Tris-HCl, 200 mM NaCl, pH7.5). After a night at -20 °C, to induce cell lysis, the cultures are sonicated (Branson Sonifier 250, 4 cycles of 50 sec, pulse at 90%, intensity at 40 arbitrary units). The cytoplasmic fraction was purified by centrifugation (18,000 × g, 30 min, 8 °C for ComK2 and SA2107, 20 °C for ComK1). The supernatant was then loaded into a Ni-NTA resin. Elutions obtained at 200 mM and/or 400 mM of Imidazole were then concentrated using ultrafiltration by centrifugation (Vivaspin protein concentrator spin columns, Cytiva). These samples were then further purified using an ÄKTA pure™ size-exclusion chromatography system (Superdex 200 column, Cytiva). Chosen fractions containing the purified ComK1 and ComK2 were then finally concentrated using ultrafitration (365 μM for ComK1, 536 μM for ComK2). To concentrate SA2107, chosen fractions were gathered, precipitated with 25% of Ammonium Sulfate, incubated overnight at 8 °C on a rotating wheel and centrifuged (18,000 x g, 1 h, 8 °C). The supernatant was then eliminated, and the pellet was resuspended in 1 mL of Tris Buffer (50 mM Tris-HCl, 200 mM NaCl, pH7.5), and dialyzed (in Phosphate Buffered Saline + Tween 0.1% solution to remove the Ammonium Sulfate) to obtain SA2107 at 50 μM. Finally, the protein samples were aliquoted in Eppendorf tubes, frozen in liquid nitrogen and stored at -80 °C.

### Flow-induced dispersion analysis (FIDA[20])
Labelled SA2107 (referred to as SA2107$_{488}$) was prepared by conjugation with ATTO 488 NHS Ester (Sigma-Aldrich). 47 μL of SA2107 (50 μM) recovered in PBS (Bio-Rad) +Tween 0.1% (Sigma) was incubated with 2.5 μL of Sodium Bicarbonate pH9 0.2 M, and 0.5 μL of ATTO 488 NHS Ester 10 mM (2-fold molar ratio of dye-to-protein) for 2 hours at room temperature, protected from light. The labelled protein was then purified using PD SpinTrap G-25 column (Cytiva), pre-equilibrated with PBS (Bio-Rad) +Tween 0.1% (Sigma) buffer to remove the free dye excess. The concentration of SA2107$_{488}$ was measured by UV-vis at 480 nm and 506 nm (absorbance of the fluorophore) using an absorption coefficient of 18 910 M-1 cm$^{-1}$.

Binding experiments were performed with a Fida 1 instrument (Fida Biosystems ApS), using laser-induced fluorescence detection with an excitation wavelength of 480 nm. A permanently coated capillary (outer diameter 375 μm, inner diameter 75 μm, length to detector 84 cm, Fida Biosystems ApS) was used in the apparatus with a controlled temperature of 25 °C.

Binding curves of SA2107$_{488}$ with ComK1 or ComK2 were obtained by using a PreMix method where the indicator sample (SA2107$_{488}$) is pre-incubated with the analyte (Non-labelled ComK1 or ComK2). To form the complexes and let the reaction reach the equilibrium state, the indicator sample is prepared with a fixed concentration of 20 nM of SA2107$_{488}$ pre-incubated for 10 minutes with ComK1 or ComK2 with a titration of 0-2 μM, in PBS (Bio-Rad) +Tween 0.1% (Sigma) buffer, in a final volume of 100 μL. The analyte sample is prepared by making a titration of 0-2 μM of ComK1 or ComK2 in PBS (Bio-Rad) +Tween 0.1% (Sigma), in a final volume of 100 μL. The capillary was first equilibrated with just the analyte sample at 3500 mbar for 30 seconds. The indicator sample is then injected at 50 mbar for 10 seconds. Finally, the indicator is mobilized and fluorescence is measured by injecting the analyte at 400 mbar for 180 seconds. Each binding experiments (SA2107$_{488}$+ComK2 or SA2107$_{488}$+ComK1) were performed thrice.

Taylorgrams obtained with this method were analyzed using Fida Software (version 3.0) with a multi-species fit to fix the ATTO 488 NHS Ester hydrodynamic radius (Rh) at 0.60 nm. This way, the apparatus is able to rigorously determine the apparent hydrodynamic radius of SA2107$_{488}$ alone (Rh), or in a complex (Rhi).

### Electromobility shift assay
To test the ability of our protein of interest (ComK1, ComK2, SA2107) to bind to the *comG* promoter, we constructed a 100 bp double-stranded probe corresponding to the *comG* operon upstream region labelled with a Cyanine5 (Cy5) fluorophore contained in one of the two primers. The promoter of *bsaA* (a gene not involved in NT) was used as a negative control. The different probes were then diluted at 1 nM in a buffer containing 20 mM of HEPES (pH 7.5, Sigma), 150 mM of NaCl and 0.01% of IGEPAL CA-630 (Sigma-Aldrich). In different Eppendorf tubes, the probes were then incubated for 15 minutes at room temperature with an increasing concentration of protein (0 to 2 μM). The samples were finally loaded on a 6% polyacrylamide gel (0.25% TBE, Bio-Rad, 5% glycerol), and migrated for 1h10min (100 V, 30 mA). The gels displaying Cy5-stained DNA were revealed in an Amersham Typhoon Biomolecular imager (Cytiva).

### Statistics and reproducibility
Dynamic expression of the *comG* operon promoter (Fig. 1a) has been repeated three times to provide mean values and standard deviations at each time point. Transformation efficiencies (Fig. 1b) were calculated based on at least 8 independent experiments (biological replicates) to calculate mean values and standard deviation for each strain. Natural transformation genes expression (Figs. 2, 4 and 5) was evaluated for each strain (mean and standard deviation) based on at least 4 independent experiments (biological replicates). Finally, FIDA experiments were conducted three times independently for each protein couple to provide mean values and standard deviations.

Results were compared (multiple comparisons to a control condition) using a two-way ANOVA with Tukey's correction (*$p < 0.05$, ***$p < 0.001$).

### Data availability
All data supporting the findings of this study are available within the paper and its Supplementary Information. Numerical source data for graphs and charts are presented in the Supplementary Data 1 file. Other details are available from the corresponding author on reasonable request.

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

## Acknowledgements

The present work has benefited from the facilities and expertise of the I2BC platform PIM supported by French Infrastructure for Integrated Structural Biology (FRISBI) ANR-10-INBS-05. We also want to thank the Imagerie-Gif flow cytometry and proteomics (SICaPS) facilities (Institute for the Integrative Biology of the Cell, I2BC, Gif sur Yvette, FRANCE) for their help and support. We thank the BIOI2 platform for making the ColabFold pipeline easily accessible at the I2BC. This work was supported by a "Young Researcher grant" from the French National Research Agency (ANR-18-CE35-0004, GenTranSa) and by the MICROBES interdisciplinary object (Paris-Saclay university) both awarded to Nicolas Mirouze. This work was also supported by PhD scholarships attributed to S. Y. Feng (Chinese Scholarship Council), Y. Arab (Paris-Saclay University scholarship) and Pierre Poirette (Fondation pour la Recherche Médicale).

## Author contributions

S.Y.F.: Conceptualization, Methodology, Investigation; Y.A.: Conceptualization, Methodology, Investigation; Y.H.: Conceptualization, Methodology, Investigation; P.P.: Conceptualization, Methodology, Investigation; M.N.: Conceptualization, Methodology, Investigation and Supervision; S.Q.-C.: Conceptualization, Methodology, Investigation, Editing, Funding acquisition and Supervision; S.M.: Conceptualization, Methodology, Investigation, Editing, Funding acquisition and Supervision; J.A.: Conceptualization, Methodology, Investigation, Editing, Funding acquisition and Supervision; N.M.: Conceptualization, Methodology, Investigation, Writing - Review & Editing, Funding acquisition and Supervision.

## Competing interests

The authors declare no competing interests.
