## [Transparent Peer Review file · Communications Biology]

ComK2 represses competence development for natural transformation in *Staphylococcus aureus* grown under anaerobic conditions

Corresponding Author: Dr Nicolas Mirouze

Version 0:

Reviewer comments:

Reviewer #1

(Remarks to the Author)

Understanding the mechanisms of gene transfer and their impact on bacterial evolution is crucial for comprehending the emergence of new bacterial clones. For many years, it was assumed that *S. aureus* is not naturally competent. However, recent reports challenge this notion. Some suggest that, in certain cases, competence may serve primarily as a mechanism for nutrient acquisition rather than gene transfer. Regardless, it is essential to explore whether competence plays any role in *S. aureus*.

The existing studies indicate that when competence does occur, the gene transfer efficiency is low. Previous work by the authors suggested that this transfer might happen under low oxygen conditions, a scenario common in *S. aureus* infections. In this report, the authors investigate competence under anaerobic conditions. However, since *S. aureus* cannot grow in such conditions, the relevance of this study is unclear. It is important to clarify in what natural environments *S. aureus* might encounter anaerobiosis while interacting with other bacterial strains, and how competence activation could benefit the bacterium, especially considering its inability to grow. A clearer explanation of this context would help in understanding the significance of the study's findings.

Other points (not in order of importance):

- The abstract dedicates too much space to previous work and should focus more on summarizing the results of this study, with a concluding sentence that explains their relevance.
- The authors should clarify the experimental setup used to test transformation under anaerobic conditions. After reading the Methods section, it is unclear whether the experiments were conducted entirely in the absence of oxygen. My understanding is that the cells were prepared in anaerobic conditions, but when mixed with media, oxygen might have been present. Please clarify this point.
- The statistical analyses for the different experiments across the figures are missing or not clearly presented.
- Regarding the experiment testing protein-protein interactions, the concentrations used seem excessively high for a regulator. Have the authors explored other methods to test this interaction in more physiologically relevant conditions? Alternatively, have the authors tried co-expressing both proteins in the same cell, tagging only one protein, and purifying the complex to confirm the interaction?

Reviewer #2

(Remarks to the Author)

The authors previously reported that *Staphylococcus aureus* is spontaneously transformable in microaerobic conditions. In this previous work, they identify SigH and ComK1 as central regulators for the control of the transformation machinery. A paralogue of ComK1, ComK2, was shown to control competence-related genes shared with the regulons of SigH and/or ComK1.

In the current manuscript, they investigate the regulation cascade leading to competence repression in anaerobic conditions. They show that ComK2 is involved in the repression of the *comG* operon (biosynthesis of competence pilus), while ComK1

and SigH are both activators of the operon. Notably, this shows that competence control in *S. aureus* relies on three different central regulators, a unique situation in Gram-positive bacteria. Independently, they identify a partner of ComK2 (i.e. SA2107) through yeast two-hybrid and biochemical assays. They propose that the complex ComK2-SA2107 acts as a repressor system of the comG operon based on simple and double knockouts. Finally, they identify the NreBC two-component system as an O₂ sensor system, which controls the repression of the comG operon potentially through the ComK2-SA2102 system.

The novelties of this work are the discovery of a role for ComK2 in competence control, the identification of a co-repressor candidate to ComK2, and a potential regulation cascade for oxygen deprivation sensing (NreBC->ComK2/SA1202->comG operon) responsible of competence repression.

The manuscript is well-written and logically organized. Although most of the data are convincing to connect the players (i.e. NreBC, ComK2, SA2102) to the control of the comG operon used as proxy of competence activation/repression, their interplay in the regulation cascade and their roles in the global regulation of competence are largely missing. Does the response regulator of the NreBC system transcriptionally control the expression of comK2 and/or sa2102? Does ComK2 directly bind to the comG promoter? Does ComK2 require SA2102 for binding? Does ComK2(-SA2102) is a competitor of ComK1 binding? What are the comK2/SA2102/NreBC regulons in anaerobic conditions?

Major comments:

1/There is no statistical analysis on the presented data as bar charts (Fig. 1b, 2, 4, and 5). Multiple comparisons to a control condition (WT) need a statistical test such as Anova 2 two-ways with Turkey post-correction. For instance, in Fig. 1b (transformation efficiency), I am not convinced that the key mutants are statistically different from the WT (very large distribution of the data) (L150-152). The authors have to find a way to improve the reproducibility of transformation assays with the WT. A second example is Fig. 2a; the comparison between WT and delta-comK2 (critical data) needs a statistical test (L152-153). Please provide an appropriate statistical analysis for all the data, update the legends, and include a statistics section in Methods.

2/L156. "... proving that only the comG operon was affected..." Does ComK2 repression in anaerobic conditions only affect the comG operon? Investigating the comK2 regulon in anaerobic conditions by RNAseq (WT vs delta-ComK2) could be interesting to identify other genes encoding the transformation machinery that could be repressed by ComK2. In the same vein, crossing the regulons of ComK2, SA2102, and NreBC could highlight their interconnection to strengthen the regulatory cascade in anaerobic conditions.

3/L175-184: NanoDSF experiments for ComK2-SA2107 interaction and L199-207: ComK1-SA2107 AlphaFold2 modelling. As AlphaFold2 is not predicting a strong complex between ComK1 and SA2107, it would have been interesting to test the ComK1-SA2107 interaction with NanoDSF as a negative control.

4/L216-221: ComK2 and SA2107 are proposed to act in the same pathway as a complex to control the expression of the comG promoter. This is only based on a genetic approach and could be strengthened by in vitro EMSA experiments on the comG promoter. Does ComK2 binding require SA2102? Does ComK2(-SA2107) compete with ComK1 for the same binding site? As most of these proteins are available, this could be attempted using protocols developed for ComK binding experiments in *Bacillus subtilis*.

5/L236-241: Except genetic evidence based on simple and double knockouts, there are no experiments showing a direct link between NreBC (i.e. NreC) and the regulatory system ComK2-SA2107. Besides the comparison of regulons as proposed above, it would be interesting to show that the response regulator of the NreBC system directly controls the expression of comK2 and/or sa2107 through the use of reporter fusions. On the other hand, it cannot be excluded that the impact of NreC inactivation leads to an increase in comK1 and/or sigH expression leading in turn to an increased expression of the comG operon.

Minor comments:

1/L162 and L191: SA2107 is a protein of unknown function without any annotated domain. Based on the predicted structure by AlphaFold2, is the fold of SA2107 found in other proteins? For example, the fold could be compared to known structures through the DALI server. Are there homologs in other species among Firmicutes? Is there any co-occurrence between SA2107 and ComK?

2/ L77: "cryptic": does SigH is cryptic anymore?

3/L70-L87-L125, ...: avoid to use too many times "importantly"

4/L108 and other titles: is it appropriate to directly refer to figures in titles?

5/L117: "shacking" - correct?

6/ L118 "competent". I would suggest removing this word because a GFP+ cell is not necessarily competent for DNA uptake

7/L163: does ref 19 is appropriate?

8/ L169-170: give a reference for NanoDSF

10/ L175-184: indicate with an arrow on panel 3b the inflexion point (Ti value) of each curve

11/ L231: "nreC mutant strain", indicates that nreC encodes the response regulator

12/ L273: "cofactor": confusing? Co-repressor? Co-modulator?

13/L497: *Escherichia coli* in italics

14/Fig3b and corresponding legend: the color code used in panel b is not correctly referred to the legend.

15/ Fig. 6b and c: arrows between SrrA-P and SigH, and NreC-P and SA2107/ComK2 should be dotted lines with question marks since there is no direct evidence of an effect on SigH and SA2107/ComK2 of the response regulators

16/ Supp Fig. 2: give accession numbers for SA2107 and ComK2

Version 1:

Reviewer comments:

Reviewer #1

(Remarks to the Author)

Thank you for your responses. However, I still find the key issues I raised insufficiently addressed. The main concern remains the biological relevance of the reported phenomenon. While you now refer to the conditions as "strong oxygen limitation" rather than "strict anaerobiosis," *Staphylococcus aureus* does not grow well under such extreme oxygen deprivation. Without a clear example of a natural scenario, such as a specific infection niche, where these extremely low oxygen conditions occur and where competence induction could offer a selective advantage, it remains unclear how your findings relate to real-life bacterial behaviour. A detailed discussion of such contexts is needed to support the biological significance of the work.

In addition, I remain unconvinced by the experimental design in which competence is induced under low oxygen but transformation is assessed under normoxic conditions. This switch undermines the interpretation of the results, as the process of natural transformation is inherently multistep and environmentally sensitive. If the transformation assay is not performed under the same conditions that trigger competence, the conclusions about oxygen's role in facilitating gene acquisition become speculative. While I appreciate the practical challenges described, the design ultimately weakens the physiological relevance of the findings.

Reviewer #2

(Remarks to the Author)

The revision of the manuscript on the function of ComK2 as a competence repressor in *Staphylococcus aureus* by Feng et al. significantly improves the strength of the data. For instance, appropriate statistical analysis of all data clarifies the relevance of the observed effects. In addition, a range of new data were obtained such as including ComK1 as a negative control in *in vitro* studies of SA2107-ComK2 binary interaction. The SA2107-ComK2 interaction was also evaluated at low concentration using an alternative technique that confirms the first results.

Concerning my specific comments, I agree with the authors that generating RNAseq data under strong oxygen limitation could be risky and will require time course experiments, which could be considered as out of the scope of the present study. Although negative results were obtained, I appreciate the attempts to clarify the regulatory cascade using reporter strains and EMSAs.

Without asking for additional experiments, I would recommend using luciferase instead of GFP as reporter for future *in vivo* assays of promoter activity, based on its much higher sensitivity (adapted to genes with low expression). For EMSA experiments shown in supplemental material, it would be interesting to add technical details (DNA probe labeling, incubation buffer, t° , ...) in the Material and Methods or in the legend of the supplemental figure.

Minor comment: Fig. 3d, please replace commas by dots.

Response to Referees

Reviewer #1 (Remarks to the Author):

We would like to thank Reviewer #1 for her/his time to review our manuscript and for the helpful comments and suggestions made. We particularly followed Reviewer #1 advice regarding the Biophysics method used in the manuscript and the low protein concentrations. We also changed the wording, using the expression “strong oxygen limitation” rather than “anaerobic conditions”.

Our answers are presented below and the changes or precisions suggested by Reviewer #1 clearly improved our manuscript.

Understanding the mechanisms of gene transfer and their impact on bacterial evolution is crucial for comprehending the emergence of new bacterial clones. For many years, it was assumed that *S. aureus* is not naturally competent. However, recent reports challenge this notion. Some suggest that, in certain cases, competence may serve primarily as a mechanism for nutrient acquisition rather than gene transfer. Regardless, it is essential to explore whether competence plays any role in *S. aureus*.

Competence for natural transformation as a mechanism for nutrient acquisition has been proposed in Gram-negative bacterial species that are ‘constitutively’ competent (i.e. *Helicobacter pylori*). This is not the case for *S. aureus* which restricts the development of competence during specific conditions. In addition, it is thought that the presence of DNA in the environment as a nutrient should itself induce the development of competence, which is not the case in *S. aureus*.

Furthermore, Morikawa and his colleagues have established that natural transformation is the only known mechanism able to transfer SCCmec, a genetic island harboring the *mecA* gene conferring resistance to methicillin (and leading to MRSA strains, <https://doi.org/10.1371/journal.ppat.1003003>). Considering the transfer of SCCmec as well as the transfer of any genetic sequences associated to a fitness advantage, we completely agree with reviewer#1 that exploring the role played by competence for natural transformation in *S. aureus* will be essential.

The existing studies indicate that when competence does occur, the gene transfer efficiency is low. Previous work by the authors suggested that this transfer might happen under low oxygen conditions, a scenario common in *S. aureus* infections. In this report, the authors investigate competence under anaerobic conditions. However, since *S. aureus* cannot grow in such conditions, the relevance of this study is unclear. It is important to clarify in what natural environments *S. aureus* might encounter anaerobiosis while interacting with other bacterial strains, and how competence activation could benefit the bacterium, especially considering its inability to grow. A clearer explanation of this context would help in understanding the significance of the study’s findings.

In the previous work mentioned by R1 (doi: 10.1038/s42003-023-04892-1), we effectively described optimized environmental conditions leading to transformation efficiencies up to 10^{-4} . This might be considered as low in respect to the 50% of competent cells. However, considering that acquisition of a sequence associated to a better fitness by a single cell is

enough to generate a problematic new clone (for example becoming MRSA through SCCmec acquisition), these transformation efficiencies might in fact be more than significant.

In this previous study (doi: 10.1038/s42003-023-04892-1) these optimized conditions were associated to oxygen rarefaction. To be more precise, the oxygen concentration dropped to around 0.3% leading to what we called microaerobic conditions. In the present report, the experimental conditions also lead to oxygen rarefaction but this time, the oxygen concentration is at least 10 times lower (i.e. below 0.03%, see Fig. 1a).

It is important to mention that the detection limit of our oxygen sensor is around 0.03% so there is a quite reasonable chance that the oxygen concentration is even lower than anticipated. We now provide a new Supp. Fig. 1 in order to display the difference in oxygen concentration in these two studies.

In order to qualify this lower oxygen concentration, we used in the first version of the manuscript the term “strict anaerobic” conditions. We agree with Reviewer #1 that this might not be appropriate. This is why we removed the mention of “strict anaerobic” conditions and replaced it by “strong oxygen limitation”.

Other points (not in order of importance):

- The abstract dedicates too much space to previous work and should focus more on summarizing the results of this study, with a concluding sentence that explains their relevance.

DONE

- The authors should clarify the experimental setup used to test transformation under anaerobic conditions. After reading the Methods section, it is unclear whether the experiments were conducted entirely in the absence of oxygen. My understanding is that the cells were prepared in anaerobic conditions, but when mixed with media, oxygen might have been present. Please clarify this point.

The purpose of this publication is to investigate the environmental conditions modulating the development of competence. Once the cells have developed competence, they are ready for natural transformation.

This is why the transformation experiments were conducted in the presence of oxygen. Addition of DNA was performed after resuspension in media with normal oxygen concentration for practical reasons. Indeed, adding DNA to the culture of competent cells requires to open the tube which will impact the oxygen concentration within the culture. In addition, the transformation protocol, as published by Morikawa (<https://doi.org/10.1371/journal.ppat.1003003>) and our laboratory (DOI: 10.1038/s42003-023-04892-1) requires the competent cells to be centrifuged and washed with fresh medium, which by definition contains normal oxygen concentrations.

However, our manuscript shows that the development of competence itself, and therefore the way *S. aureus* cells sense their environment and the oxygen concentration, switch on specific pathways to modulate the percentage of cells inducing the development of competence.

In addition, the transformation experiments, even though performed in the presence of oxygen, perfectly match the percentage of cells inducing competence (i.e. less competent cells implies lower transformation efficiencies with adequate ratios).

We have updated the Methods section to be sure that there is no misunderstanding.

- The statistical analyses for the different experiments across the figures are missing or not clearly presented.

We have globally updated the statistical analysis in the manuscript. We performed a two-way ANOVA with Tukey's posthoc test (as proposed by Reviewer #2) to evaluate how significant is the difference between two groups of independent data (see Fig. 1b, 2, 4 and 5).

We also increased the number of measurements and improved our reproducibility.

We also updated the Figures legend and now include a statistical analysis section in the Methods.

- Regarding the experiment testing protein-protein interactions, the concentrations used seem excessively high for a regulator. Have the authors explored other methods to test this

interaction in more physiologically relevant conditions? Alternatively, have the authors tried co-expressing both proteins in the same cell, tagging only one protein, and purifying the complex to confirm the interaction?

We agree with Reviewer #1 that being able to test lower protein concentrations could be important to reach more physiologically relevant conditions. Unfortunately, the physical principle of nanoDSF does not allow to decrease the proteins concentration. This is why, as proposed by Reviewer #1, we decided to use another method named Flow-Induced Dispersion Analysis (FIDA). FIDA is based on measuring the change in size of a ligand as it selectively interacts with a target protein. The K_d of the interaction may be obtained in a titration experiment allowing us to considerably decrease the proteins concentration.

In our experiments, we labeled SA2107 and determined by FIDA its average radius around 1.97 ± 0.024 nm. When exposed to ComK2, the measured radius increased to 2.75 ± 0.09 nm as expected when SA2107 and ComK2 interact. All these numbers are coherent with the predictions made by AlphaFold for the SA2107 radius (2.05 ± 0.024 nm) as well as for the complex (evaluated around 2.81 ± 0.09 nm).

Importantly, using this method we could test protein concentrations 10 to 1000 times lower than previously and evaluate a K_d around 420 nM for the SA2107/ComK2 interaction.

Finally, as proposed by Reviewer #2, we also performed a control experiment with ComK1, as AlphaFold predicts that SA2107 specifically interacts with ComK2. This is why we purified *S. aureus* ComK1 from *E. coli* and confirmed that SA2107 radius does not change when exposed to ComK1.

Reviewer #2 (Remarks to the Author):

We would like to thank Reviewer #2 for her/his time to review our manuscript and for the helpful comments and suggestions made. We particularly followed Reviewer #2 advice regarding the use of purified ComK1 as a negative control in the Biophysics experiments. We also attempted to characterize the complete regulatory pathway.

Our answers are presented below and the changes or precisions suggested by Reviewer #2 clearly improved our manuscript.

The authors previously reported that *Staphylococcus aureus* is spontaneously transformable in microaerobic conditions. In this previous work, they identify SigH and ComK1 as central regulators for the control of the transformation machinery. A paralogue of ComK1, ComK2, was shown to control competence-related genes shared with the regulons of SigH and/or ComK1.

In the current manuscript, they investigate the regulation cascade leading to competence repression in anaerobic conditions. They show that ComK2 is involved in the repression of the comG operon (biosynthesis of competence pilus), while ComK1 and SigH are both activators of the operon. Notably, this shows that competence control in *S. aureus* relies on three different central regulators, a unique situation in Gram-positive bacteria. Independently, they identify a partner of ComK2 (i.e. SA2107) through yeast two-hybrid and biochemical assays. They propose that the complex ComK2-SA2107 acts as a repressor system of the comG operon based on simple and double knockouts. Finally, they identify the NreBC two-component system as an O₂ sensor system, which controls the repression of the comG operon potentially through the ComK2-SA2102 system.

The novelties of this work are the discovery of a role for ComK2 in competence control, the identification of a co-repressor candidate to ComK2, and a potential regulation cascade for oxygen deprivation sensing (NreBC->ComK2/SA2102->comG operon) responsible of competence repression.

The manuscript is well-written and logically organized. Although most of the data are convincing to connect the players (i.e. NreBC, ComK2, SA2102) to the control of the comG operon used as proxy of competence activation/repression, their interplay in the regulation cascade and their roles in the global regulation of competence are largely missing. Does the response regulator of the NreBC system transcriptionally control the expression of comK2 and/or sa2102? Does ComK2 directly bind to the comG promoter? Does ComK2 require SA2102 for binding? Does ComK2(-SA2102) is a competitor of ComK1 binding? What are the comK2/SA2102/NreBC regulons in anaerobic conditions?

Major comments:

1/There is no statistical analysis on the presented data as bar charts (Fig. 1b, 2, 4, and 5). Multiple comparisons to a control condition (WT) need a statistical test such as Anova 2 two-ways with Turkey post-correction. For instance, in Fig. 1b (transformation efficiency), I am not convinced that the key mutants are statistically different from the WT (very large distribution of the data) (L150-152). The authors have to find a way to improve the reproducibility of transformation assays with the WT. A second example is Fig. 2a; the comparison between WT and delta-comK2 (critical data) needs a statistical test (L152-153). Please provide an appropriate statistical analysis for all the data, update the legends, and include a statistics section in Methods.

We have globally updated the statistical analysis in the manuscript. We performed a two-way ANOVA with Tukey's posthoc test (as proposed by Reviewer #2) to evaluate how significant is the difference between two groups of independent data (see Fig. 1b, 2, 4 and 5).

We also increased the number of measurements and improved our reproducibility.

We also updated the Figures legend and now include a statistical analysis section in the Methods.

2/L156. "... proving that only the comG operon was affected..." Does ComK2 repression in anaerobic conditions only affect the comG operon? Investigating the comK2 regulon in anaerobic conditions by RNAseq (WT vs delta-ComK2) could be interesting to identify other genes encoding the transformation machinery that could be repressed by ComK2. In the same vein, crossing the regulons of ComK2, SA2102, and NreBC could highlight their interconnection to strengthen the regulatory cascade in anaerobic conditions.

The *comG* operon is essential for natural transformation. As a consequence, a lower expression of this operon will be enough to limit natural transformation, independently from the expression of other natural transformation genes expression.

Even though we agree with Reviewer #2 that having a complete picture of the impact of ComK2 under anaerobic conditions would be interesting, we think that this kind of approach is high risk.

Indeed, GFP stability allowed us to wait for the end of the experiment and evaluate the percentage of GFP-positive competent cells that appeared throughout growth. With this kind of approach, we found 4 times more competent cells in *comK2*, *sa2107* and *nreC* mutant strains than in wild type.

In contrast, RNA-seq gives a picture of the content of total mRNA at a specific time. Therefore, (i) we would have to choose the right time at which a difference can be observed, which might be different depending on the genes considered and (ii) the difference observed between mutant and wt strains at a specific time point might not be as significant as observed using the GFP accumulated throughout the experiment.

Therefore, we feel that such approach would be high risk for the present project that already shows a lot of data, but we will plan to realize such experiments in the future.

3/L175-184: NanoDSF experiments for ComK2-SA2107 interaction and L199-207: ComK1-

SA2107 AlphaFold2 modelling. As AlphaFold2 is not predicting a strong complex between ComK1 and SA2107, it would have been interesting to test the ComK1-SA2107 interaction with NanoDSF as a negative control.

As explained above in the answer to Reviewer #1, we now use a different method named Flow-Induced Dispersion Analysis (FIDA). This new method allowed us to compare the radii of SA2107 alone and of the complex between SA2107 and ComK2 with the same radii estimated by AlphaFold. Importantly, this method allowed us to confirm this interaction, using more physiologically relevant concentrations.

Since we agree with Reviewer #2 that using ComK1 would be an important negative control, we purified the ComK1 regulator from *E. coli* and tested by FIDA the ability of ComK1 to interact with SA2107. As expected, the radius of SA2107 did not change in the presence of important concentrations of ComK1, proving that these two proteins do not interact.

4/L216-221: ComK2 and SA2107 are proposed to act in the same pathway as a complex to control the expression of the comG promoter. This is only based on a genetic approach and could be strengthened by in vitro EMSA experiments on the comG promoter. Does ComK2 binding require SA2102? Does ComK2(-SA2107) compete with ComK1 for the same binding site? As most of these proteins are available, this could be attempted using protocols developed for ComK binding experiments in Bacillus subtilis.

We agree with Reviewer #2 that *in vitro* EMSA experiments could strengthen our hypotheses. This is why we tested the ability of SA2107 alone, ComK2 alone or the complex between SA2107 and ComK2 to specifically bind to the *comG* promoter (see new Supp. Fig. 8). Unfortunately, we could not detect any shift and cannot conclude on the ability of this complex to repress the *comG* operon expression by directly binding to its promoter.

We also wondered if the SA2107/ComK2 complex could prevent ComK1 to access the *comG* promoter. Unfortunately, when we tested ComK1 alone, we observed an important unspecific DNA binding, even on the control DNA (promoter of a gene not controlled by ComK1 in previous RNAseq experiments).

Therefore, despite our efforts to strengthen our hypotheses, the experiments realized do not allow us to make stronger conclusions.

5/L236-241: Except genetic evidence based on simple and double knockouts, there are no experiments showing a direct link between NreBC (i.e. NreC) and the regulatory system ComK2-SA2107. Besides the comparison of regulons as proposed above, it would be interesting to show that the response regulator of the NreBC system directly controls the expression of comK2 and/or sa2107 through the use of reporter fusions. On the other hand, it cannot be excluded that the impact of NreC inactivation leads to an increase in comK1 and/or sigH expression leading in turn to an increased expression of the comG operon.

We agree with Reviewer #2 that showing that NreBC directly controls *sa2107* expression would be an important experiment to strengthen our hypothesis of a single regulatory pathway. This is why we created a new P_{sa2107} -gfp transcriptional fusion and measured the percentage of GFP-expressing cells in wild type and *nreC* mutant strains. Unfortunately, we could not detect

any GFP-positive cells probably showing that *sa2107* expression is too low and below our detection limit.

Minor comments:

1/L162 and L191: SA2107 is a protein of unknown function without any annotated domain. Based on the predicted structure by AlphaFold2, is the fold of SA2107 found in other proteins? For example, the fold could be compared to known structures through the DALI server. Are there homologs in other species among Firmicutes? Is there any co-occurrence between SA2107 and ComK?

This is a very interesting question. While SA2107 has no conserved domain, its predicted general structure is very similar to the fold of BssR (also called YliH) from *E. coli* found through Foldseek search in the AlphaFold database (AFDB) proteome. Importantly, BssR's name comes from the phrase "regulator of biofilm through signal secretion". Indeed, BssR is a small regulator involved in the inhibition of biofilm formation, a function similar to SA2107 during competence under near-anaerobic conditions. We now mention this in the discussion lines 286-292.

We also looked for any co-occurrence between *sa2107* and *comK* among firmicutes. Unfortunately, *sa2107* is only found in *Staphylococcus aureus* species.

2/ L77: "cryptic": does SigH is cryptic anymore?

We removed the term cryptic.

3/L70-L87-L125, ...: avoid to use too many times "importantly"

Done

4/L108 and other titles: is it appropriate to directly refer to figures in titles?

We usually provide the figures for the reviewers to facilitate the manuscript reading for referees. We removed them now.

5/L117: "shacking" - correct?

Replaced by statically

6/ L118 "competent". I would suggest removing this word because a GFP+ cell is not necessarily competent for DNA uptake

Done

7/L163: does ref 19 is appropriate?

In ref 19 "Mortier-Barrière et al. Cell. 2007", Fig. 5 shows the interaction between DprA (used as a bait) and a fragment of RecA (deleted of its N-terminal end) from a *S. pneumoniae* genomic library in yeast.

8/ L169-170: give a reference for NanoDSF

We do not show NanoDSF experiments anymore. We replace by Flow-Induced Dispersion Analysis (FIDA) and provide the adequate Reference.

10/ L175-184: indicate with an arrow on panel 3b the inflexion point (Ti value) of each curve

We changed the method.

11/ L231: “nreC mutant strain”, indicates that nreC encodes the response regulator

Done

12/ L273: “cofactor”: confusing? Co-repressor? Co-modulator?

Done

13/L497: Escherichia coli in italics

Done

14/ Fig3b and corresponding legend: the color code used in panel b is not correctly referred to the legend.

We now provide a new Fig. 3b (new experiment based on new technology: FIDA).

15/ Fig. 6b and c: arrows between SrrA-P and SigH, and NreC-P and SA2107/ComK2 should be dotted lines with question marks since there is no direct evidence of an effect on SigH and SA2107/ComK2 of the response regulators

Done

16/ Supp Fig. 2: give accession numbers for SA2107 and ComK2

Done

Response to Referees 2nd round

Reviewer #1 (Remarks to the Author):

Thank you for your responses. However, I still find the key issues I raised insufficiently addressed. The main concern remains the biological relevance of the reported phenomenon. While you now refer to the conditions as “strong oxygen limitation” rather than “strict anaerobiosis,” *Staphylococcus aureus* does not grow well under such extreme oxygen deprivation. Without a clear example of a natural scenario, such as a specific infection niche, where these extremely low oxygen conditions occur and where competence induction could offer a selective advantage, it remains unclear how your findings relate to real-life bacterial behaviour. A detailed discussion of such contexts is needed to support the biological significance of the work.

The message of the publication is that competence and NT are limited under such oxygen-limited conditions and we present evidence that this is regulated by a dedicated pathway involving an oxygen-sensing TCS. We are not saying that such strong oxygen limitation offers a selective advantage but rather the opposite: *S. aureus* requires more oxygen (at least 10 times more) for competence and NT to be optimal.

However, we have no problem to discuss the physiological/biological relevance of such conditions in vivo. We added this paragraph in the discussion lines 322-336 where we particularly focus on infected wounds, cystic fibrosis and biofilm formation.

As pointed out by R1, we also mention that under these conditions, growth is limited which would even further decrease *S. aureus* ability to validate any new genetic acquisition through natural transformation (lines 342-344).

In addition, I remain unconvinced by the experimental design in which competence is induced under low oxygen but transformation is assessed under normoxic conditions. This switch undermines the interpretation of the results, as the process of natural transformation is inherently multistep and environmentally sensitive. If the transformation assay is not performed under the same conditions that trigger competence, the conclusions about oxygen’s role in facilitating gene acquisition become speculative. While I appreciate the practical challenges described, the design ultimately weakens the physiological relevance of the findings.

We completely agree with R1 that, for technical limitations, our natural transformation assay is not perfect. We now make clear in the text lines 134-140 that this is a limit of our assay.

However, we also mention that the decrease in the percentage of competence cells under strong limitation is in perfect agreement with the decrease observed in the natural transformation efficiency. We also wrote that we cannot exclude the fact that new competent cells could appear after the addition of DNA but in that case, it means that we underestimate the limitation of competence development that is at the center of this publication.

Reviewer #2 (Remarks to the Author):

The revision of the manuscript on the function of ComK2 as a competence repressor in *Staphylococcus aureus* by Feng et al. significantly improves the strength of the data. For instance, appropriate statistical analysis of all data clarifies the relevance of the observed effects. In addition, a range of new data were obtained such as including ComK1 as a negative control in in vitro studies of SA2107-ComK2 binary interaction. The SA2107-ComK2 interaction was also evaluated at low concentration using an alternative technique that confirms the first results.

Concerning my specific comments, I agree with the authors that generating RNAseq data under strong oxygen limitation could be risky and will require time course experiments, which could be considered as out of the scope of the present study. Although negative results were obtained, I appreciate the attempts to clarify the regulatory cascade using reporter strains and EMSAs.

We appreciate that R2 noted the attempts to clarify the regulatory cascade. We will for sure continue to investigate the links, whether they are direct or indirect, articulating NreBC, SA2107, ComK2 and the natural transformation genes expression.

Without asking for additional experiments, I would recommend using luciferase instead of GFP as reporter for future in vivo assays of promoter activity, based on its much higher sensitivity (adapted to genes with low expression).

We completely agree with R2 that the use of the luciferase allows a much higher sensitivity which could make a difference in the detection of low-expressed genes. We have been working on the system in our laboratory and hope to improve our sensibility.

For EMSA experiments shown in supplemental material, it would be interesting to add technical details (DNA probe labeling, incubation buffer, t°, ...) in the Material and Methods or in the legend of the supplemental figure.

Done. See lines 360-639

Minor comment: Fig. 3d, please replace commas by dots.

Done